# Taxonomic and Phylogenetic Studies of Two Brackish *Pleuronema* Species (Protista, Ciliophora, Scuticociliatia) from Subtropical Coastal Waters of China, with Report of a New Species

**DOI:** 10.3390/microorganisms11061422

**Published:** 2023-05-27

**Authors:** Hui Zhang, Xuetong Zhao, Tingting Ye, Zehao Wu, Fan Wu, Xiangrui Chen, Mingjian Liu

**Affiliations:** 1School of Marine Sciences, Ningbo University, Ningbo 315800, China; zzhang_hui@163.com (H.Z.); zhaoxuetong1102@163.com (X.Z.); yetingting000@126.com (T.Y.); wuzehao@126.com (Z.W.); yueguil@126.com (F.W.); 2Laboratory of Evolution and Marine Biodiversity, Institute of Evolution and Marine Biodiversity & College of Marine Life Sciences, Ocean University of China, Qingdao 266003, China

**Keywords:** ciliates, morphology, new species, pleuronematids, SSU rDNA phylogeny

## Abstract

The genus *Pleuronema* Dujardin, 1841, with nearly 40 morphospecies, is one of the largest genera in the well-known subclass Scuticociliatia. In the present study, two *Pleuronema* species were collected from subtropical coastal waters of the East China Sea. The morphology and molecular phylogeny were investigated using modern standard methods. *Pleuronema ningboensis* n. sp. is mainly characterized by an elliptical body in outline with the right ventrolateral side straight, 16–22 somatic kineties, 3–5 preoral kineties, and the posterior end of the membranelle 2a hook-like. An improved diagnosis of *Pleuronema orientale* Pan et al., 2015 was provided: body size in vivo usually 90–135 × 45–85 μm, right ventrolateral side convex, 36–51 somatic kineties, 1–5 preoral kineties, one to three spherical macronuclei, membranelle 2a arranged in a zig-zag pattern in middle portion, posterior region hook-like, both membranelle 1 and membranelle 3 composed of three rows of basal bodies. The small subunit ribosomal DNA (SSU rDNA) of two species is sequenced, and their molecular phylogeny is analyzed. The new species *Pleuronema ningboensis* n. sp. clusters with *P. grolierei* KF840519, *P. setigerum* JX310015, *P. paucisaetosum* KF206430, and *P.* cf. *setigerum* KF848875, basically in accord with the morphological characteristics.

## 1. Introduction

Ciliated protozoa (ciliates) are an extremely diverse and ubiquitous group with more than 10,000 nominal species from a wide range of habitats [1]. They are considered a major link in microbial food webs and play an important role in energy flow and material circulation in aquatic environments [2,3,4,5]. The subclass Scuticociliatia is a speciose assemblage of ciliates that are found in various aquatic and terrestrial habitats [6,7]. Some scuticociliates are well known for their facultative parasitism and cause serious diseases of aquatic organisms [8,9,10,11]. Actually, most ciliates assigned to this subclass are free-living with significant ecological functions regarding their nutrition style, behavior, and many other biological characteristics [12,13,14,15,16].

Pleuronematida is one of four orders within the subclass Scuticociliatia and most Pleuronematida species are free-living [1]. They are mainly characterized by an expansive oral region with a prominent, curtain-like paroral membrane [17]. As the representative genus of this order, *Pleuronema* is a speciose and cosmopolitan genus comprising nearly 40 known morphospecies which can be found in various aquatic environments and mainly distributed in the intertidal zone [18,19,20,21,22,23]. In the past decade, about 10 new species have been discovered, indicating a high species diversity of this genus, and a further study on the biodiversity of this genus remains to be explored [23,24,25,26].

In the present work, two *Pleuronema* species, *Pleuronema ningboensis* n. sp. and *P. orientale* Pan et al., 2015, were collected from the subtropical coastal wetlands in China. They were investigated both in vivo and by using silver staining methods. In addition, phylogenetic analysis was performed based on the SSU rDNA sequences to reveal the evolutionary relationships of Pleuronematidae.

## 2. Materials and Methods

### 2.1. Sample Collection, Observation, and Identification

Samples of subtropical brackish waters were collected from coastal wetlands in Ningbo, China (Figure 1). *Pleuronema ningboensis* n. sp. was collected on 13 November 2019 from an intertidal sand (29°46′33′′ N, 121°57′47′′ E). The water temperature was 19 °C, and the salinity was 19‰. *Pleuronema orientale* was collected on 7 May 2020 from a ditch on Meishan Island (29°77′63′′ N, 121°95′20′′ E). The water temperature was 24 °C, and the salinity was 4.5‰. For *Pleuronema ningboensis* n. sp., several holes (diameter and depth were about 10 cm) were excavated on the beach when at low tide. After water gradually seeped into the holes, water and sand at the bottom of the holes were mixed and collected. An approximately 400 mL water sample was placed into a 500 mL bottle. For *P. orientale*, approximately 400 mL of well-stirred brackish water sample with humus was placed into a 500 mL bottle. Samples were then maintained in the laboratory for about one week as raw cultures in Petri dishes, and individuals were directly isolated using micropipettes for live observation and protargol staining.

Living cells were observed under a bright field and differential interference contrast microscopy (Leica DM2500) at 100–1000× magnification. The ciliature and nuclear apparatus were revealed with the protargol staining method [27]. Counts, measurements, and drawings of stained specimens were performed at 1000× magnification. Terminology and systematics follow Lynn [1] and Gao et al. [28], respectively.

### 2.2. DNA Extraction, PCR Amplification, and Gene Sequencing

For each species, five to ten cells were selected from pure cultures and washed five times with filtered in situ water (0.22 μm, Millex-GP filter unit) to exclude contamination. Then cells were placed into three Eppendorf tubes with one and multiple individuals. Genomic DNA was extracted using the DNeasy Blood and Tissue Kit (Qiagen, Hilden, Germany) following the optimized manufacturer’s protocol. The SSU rDNA was amplified using the primers 18S-F (5′-AAC CTG GTT GAT CCT GCC AGT-3′) and 18S-R (5′-TGA TCC TTC TGC AGG TTC ACC TAC-3′) [29].

**Figure 1 microorganisms-11-01422-f001:**
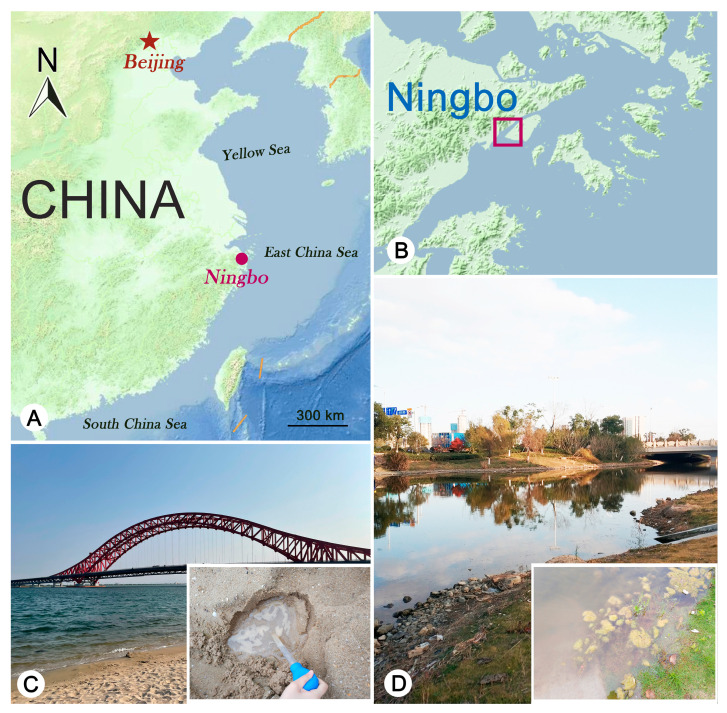
Maps showing the sampling locations in Ningbo and photographs of the sampling sites. (**A**) Portion of the map of China showing the location of Ningbo. (**B**) Map showing the sampling locations in Meishan Wetlands (colored square indicates the sampling sites). (**C**) Sampling site for *Pleuronema ningboensis* n. sp. (**D**) Sampling site for *Pleuronema orientale*.

### 2.3. Phylogenetic Analyses

Two new SSU rDNA sequences from the present work and 33 reference sequences obtained from the National Center for Biotechnology Information (NCBI) GenBank database were selected for the phylogenetic analyses. The accession number of each sequence was provided to the right of species names in the phylogenetic tree (also listed in Appendix A). Four Histiobalantiidae, namely *Falcicyclidium plouneouri* (FJ868181), *Falcicyclidium fangi* (FJ868185), *Acucyclidium atractodes* (FJ868182), *Hippocomos salinus* (JX310012), and one Eurystomatellidae, *Wilbertia typica* (FJ490551) were selected as out-group species (Appendix A). A total of 35 sequences were aligned using MUSCLE implemented in GUIDANCE with default parameters [30]. The ends of the resulting alignments were trimmed manually using the program BioEdit v7.0.5.3 [31]. Maximum likelihood (ML) analysis was conducted on the CIPRES Science Gateway server, using the RAxML-HPC2 located on XSEDE v8.2.9 [32], with the GTR + I + G model. Support for the best ML tree was calculated from 1000 bootstrap replicates. Bayesian inference (BI) analysis was performed on CIPRES Science Gateway with MrBayes on XSEDE v3.2.7a [33] using the GTR + I + G model selected by MrModeltest v2.2 [34]. The BI analysis was run for 10^6^ generations, with trees sampled every 100th generation, with the first 2500 trees discarded as burn-in. The tree topologies were visualized via MEGA v7.0 [35] and TreeView v.1.6.6 [36].

## 3. Results

Subclass Scuticociliatia Small, 1967

Order Pleuronematida Fauré-Fremiet in Corliss, 1956

Family Pleuronematidae Kent, 1881

Genus *Pleuronema* Dujardin, 1841

### 3.1. Pleuronema ningboensis n. *sp.*

#### 3.1.1. Diagnosis

Cell size in vivo about 55–65 × 25–30 μm. Right ventrolateral side straight. 16–22 somatic kineties and three to five preoral kineties. Single macronucleus. Membranelle 2a two-rowed in anterior and posterior portions, single-rowed in mid-portion, posterior end hook-like; membranelle 3 three-rowed, leftmost row shortened and posteriorly located, rightmost row slightly shortened posteriorly. Brackish water habitat.

#### 3.1.2. Type Locality

A sandy beach in the intertidal zone near Ningbo (29°77′63′′ N, 121°95′20′′ E), China. The water temperature was 19 °C, and the salinity was 19‰.

#### 3.1.3. Etymology

This species-group name, *ningboensis,* refers to the area (Ningbo, China) where the sample was collected.

#### 3.1.4. Type Materials

Two slides with protargol-stained specimens have been deposited in the Laboratory of Protozoology, Ningbo University, including one slide (registration number: YTT-20191113-2-1) with the holotype specimen circled in black ink and one slide (registration number: YTT-20191113-2-2) with paratype specimens. Another two slides with protargol-stained paratype specimens (registration number: YTT-20191113-2-3, 2-4) have been deposited to the Laboratory of Protozoology, Ocean University of China.

#### 3.1.5. Morphological Description

Living cell approximately 55–65 × 25–30 μm, outline elliptical; anterior and posterior ends broadly rounded; right ventrolateral side straight (Figure 2A and Figure 3A–C). Buccal field broad, occupying about 75% of body length. Oral cilia approximately 25 μm long. Pellicle slightly notched with shallow longitudinal grooves (Figure 3A), beneath which extrusomes closely arranged, about 2 μm in length (Figure 3D). Cytoplasm transparent to greyish, containing several food vacuoles, refractile globules, and crystals usually concentrated in posterior of cell (Figure 3E). Single contractile vacuole dorsally located about 80% down length of cell, approximately 12–15 μm in diameter when fully expanded, pulsating at interval of about 35–40 s (Figure 3C). One spherical to ellipsoidal macronucleus positioned in body center slightly anteriorly (Figure 3B), diameter of about 24 μm after protargol staining (Figure 3J, Table 1). Somatic cilia densely packed, about 9 μm long. Ten to seventeen prolonged caudal cilia, each about 20 μm in length, projecting radially from posterior end of cell (Figure 2A and Figure 3E). Movement moderately fast, rotating around main body axis and occasionally resting on substrate for some time.

**Figure 2 microorganisms-11-01422-f002:**
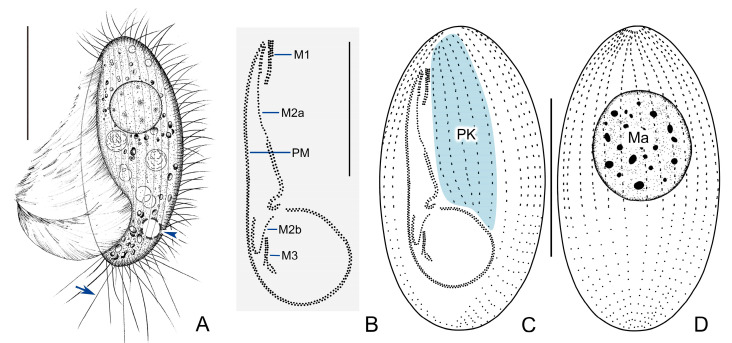
*Pleuronema ningboensis* n. sp. in vivo (**A**) and after protargol staining (**B**–**D**). (**A**) Left ventral view of a typical individual; arrowhead shows the contractile vacuole, arrow points to caudal cilia. (**B**) Detail of oral apparatus. (**C**,**D**) Left ventrolateral (**C**) and right dorsolateral (**D**) views to show the ciliature. M1, membranelle 1; M2a, membranelle 2a; M2b, membranelle 2b; M3, membranelle 3; PM, paroral membrane; PK, paroral kineties; Ma, macronucleus. Scale bars = 30 μm (**A**,**C**,**D**); 20 μm (**B**).

**Figure 3 microorganisms-11-01422-f003:**
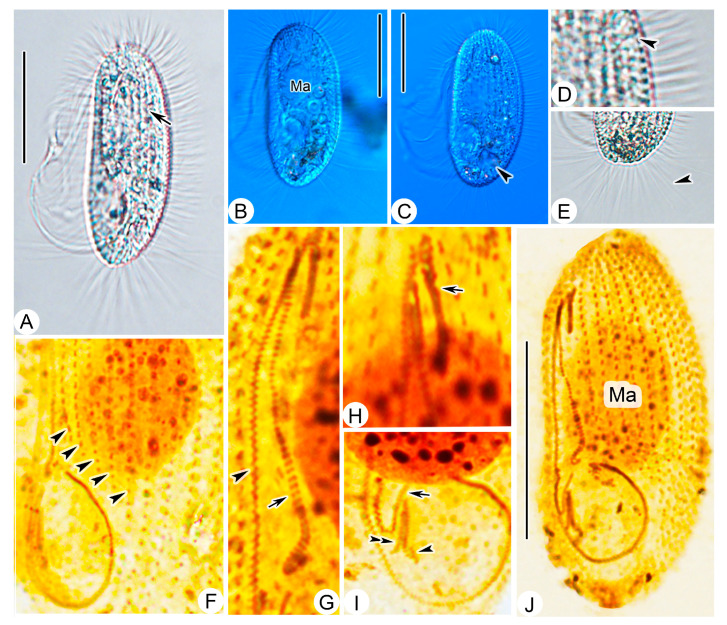
*Pleuronema ningboensis* n. sp. from life (**A**–**E**) and after protargol impregnation (**F**–**J**). (**A**) Left ventrolateral view of a typical individual in vivo; arrow indicates the grooves on the cell surface. (**B**,**C**) different individuals, showing the variation in cell view; arrowhead in (**C**) indicates the contractile vacuole. (**D**) Detail of the cortex; arrowhead indicates extrusomes. (**E**) Posterior portion of cell; arrowhead points to caudal cilia. (**F**) Detail of ciliature, arrowheads mark the posterior end of preoral kineties. (**G**) Anterior portion of the oral apparatus, arrowhead points to paroral membrane, arrow points to membranelle 2a. (**H**) To show membranelle 1 (arrow). (**I**) Posterior portion of the oral apparatus, to show membranelle 2b (arrow) and membranelle 3 (arrowhead indicates the leftmost row, double arrowheads indicate the rightmost row). (**J**) Left ventral view of the holotype specimen. Ma, macronucleus. Scale bars = 30 μm.

Sixteen to twenty-two somatic kineties (SK) extending entire length of cell to form small, bald apical plate at anterior end of cell (Figure 2D, Table 1). Each with densely spaced dikinetids in anterior 60% and monokinetids in posterior 40% (Figure 2C,D, and Figure 3J). Three to five preoral kineties (PK) located to left of buccal field, commencing near anterior end of cell and terminating posteriorly 60–70% down length of cell (Figure 2C and Figure 3F, Table 1).

Membranelle 1 (M1) three-rowed in anterior 35–40% portion, two-rowed in remaining portion (Figure 2B and Figure 3H); anterior 15% and posterior 40% of membranelle 2a (M2a) double-rowed, and middle portion single-rowed. Posterior portion of M2a hook-like, not closed into ring-like structure (Figure 2B and Figure 3G,J). Membranelle 2b (M2b) basically V-shaped, separately from M2a; left portion of M2b with basal bodies arranged in single row, and right portion arranged in zig-zag pattern (Figure 2B and Figure 3I). Membranelle 3 (M3) composed of three rows of basal bodies, leftmost row shortened and posteriorly located, with length of approximately 25% of two right rows (Figure 2B and Figure 3I). Paroral membrane (PM) prominent, and basal bodies in zig-zag pattern, occupying 75% of body length (Figure 2B,C and Figure 3G). Micronucleus not detected.

#### 3.1.6. SSU rDNA Sequence

The sequence of *Pleuronema ningboensis* n. sp. was deposited in GenBank with accession number OQ591738. The length and G + C content of the sequence are 1658 bp and 43.00%.

### 3.2. Pleuronema orientale Pan et al., 2015

This species was originally collected from Hangzhou Bay, China (30°35′33.8′′ N, 122°04′59.1′′ E) and detailed described by Pan et al. [24] based on living cells and protargol staining specimens. As a new contribution, here we describe a subtropical population and provide some new information.

#### 3.2.1. Improved Diagnosis

Body size in vivo usually 90–135 × 45–85 μm. Right ventrolateral side convex. 36–51 somatic kineties and 1–5 preoral kineties. Usually one spherical macronucleus. Membranelle 2a arranged in a zig-zag pattern in middle portion, posterior region hook-like. Brackish water habitat.

#### 3.2.2. Voucher Slides

Three voucher slides containing protargol-stained specimens have been deposited in the Laboratory of Protozoology, Ningbo University (registration number: YTT-20200507-1, 2, 3).

#### 3.2.3. Morphological Description of Ningbo Population

Living cell about 90–105 × 45–60 μm, elliptical or oval in outline. Anterior end slightly narrowed in some individuals, posterior end rounded, right ventrolateral and dorsal sides convex (Figure 4A and Figure 5A–C). Buccal field occupies 75% of body length, PM prominent (Figure 4A and Figure 5A,D). Pellicle rigid and slightly notched, beneath which extrusomes closely arranged, about 6 μm in length (Figure 4A). Cytoplasm usually colorless and transparent, containing numerous food vacuoles, refractile globules, and crystals that are usually ventrally distributed (Figure 4A and Figure 5A,B). Single contractile vacuole dorsally located approximately 80% down length of body, about 10 μm across when fully expanded, pulsating at interval of about 20–40 s (Figure 5B,C). One to three macronuclei (usually one spherical macronucleus, in 23 out of 25 cells examined), diameter approximately 32 μm after protargol staining (Figure 4C–E,G and Figure 5F,K, Table 1). Micronucleus not detected. Somatic cilia densely packed, about 10 μm in length; 10–13 prolonged caudal cilia, each 20–25 μm in length (Figure 4A and Figure 5A). Locomotion moderately fast while rotating about main body axis, sometimes drifting or lying motionless on debris for short periods.

Thirty-six to fifty-one somatic kineties extending over entire length of cell to form small, bald apical plate at anterior end of body (Figure 4G, Table 1). Kineties consist of dikinetids in anterior 75% of cell and monokinetids in posterior 25% (Figure 4F,G and Figure 5K). One to five preoral kineties located to left of buccal field, commencing near anterior end of cell and terminating posteriorly 75–80% down length of cell (Figure 4F and Figure 5E,H, Table 1).

Anterior 20% of M1 three-rowed while rest two-rowed (Figure 4B and Figure 5G). Anterior 20% and posterior 35% of M2a two-rowed, basal bodies in mid-portion arranged in zig-zag pattern; posterior portion of M2a hook-like (Figure 4B and Figure 5J); M2b V-shaped, basal bodies arranged in several single-rowed groups, each group composed of about six basal bodies (Figure 4B and Figure 5J). M3 three-rowed and closely packed, posterior end of rightmost row slightly separated from other rows by diverging rightwards (Figure 4B and Figure 5I). PM prominent, with basal bodies arranged in zig-zag pattern, occupying 75% of body length (Figure 4A,B,F, and Figure 5J).

**Figure 4 microorganisms-11-01422-f004:**
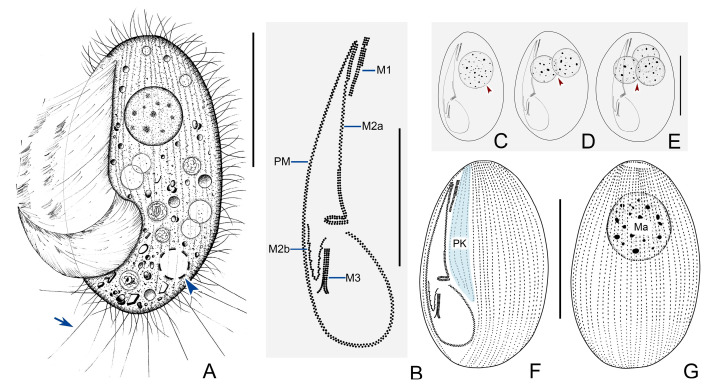
*Pleuronema orientale* from life (**A**) and after protargol impregnation (**B**–**G**). (**A**) Left ventral view of a representative individual, arrow indicates caudal cilia; arrowhead indicates contractile vacuole. (**B**) Detail of the oral apparatus. (**C**–**E**) To show the different numbers of macronuclei in different individuals; arrowheads indicate macronuclei. (**F**,**G**) Left ventrolateral (**F**) and right dorsolateral (**G**) views to show ciliature and macronucleus. M1, membranelle 1; M2a, membranelle 2a; M2b, membranelle 2b; M3, membranelle 3; PM, paroral membrane; PK, paroral kineties; Ma, macronucleus. Scale bars = 50 μm (**A**,**C**–**G**); 30 μm (**B**).

**Figure 5 microorganisms-11-01422-f005:**
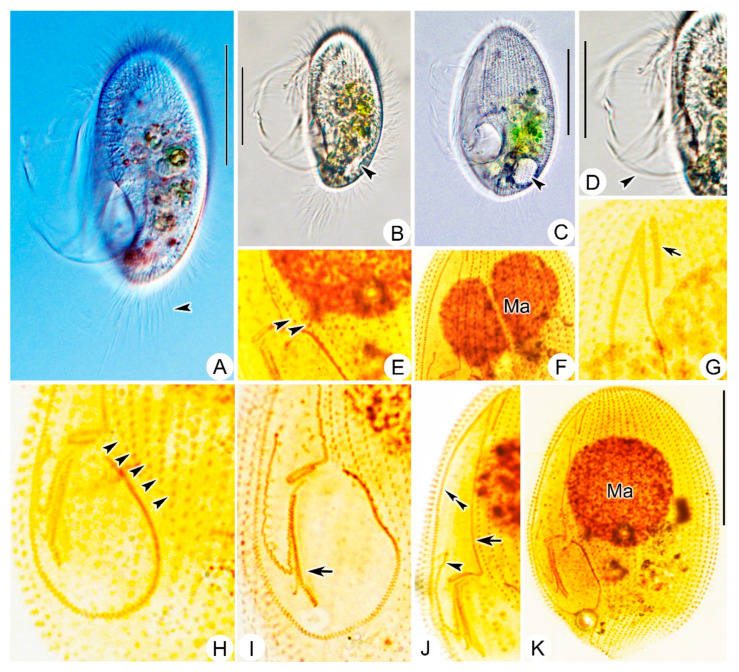
Photomicrographs of *Pleuronema orientale* in vivo (**A**–**D**) and after protargol staining (**E**–**K**). (**A**–**C**) Left ventrolateral views of different individuals; arrowhead in (**A**) marks caudal cilia, arrowheads in (**B**,**C**) point to contractile vacuole. (**D**) To show the detail of paroral membrane (arrowhead). (**E**,**H**) To show different numbers of preoral kineties (arrowheads). (**F**) Detail of double macronuclei. (**G**) To show membranelle 1 (arrow). (**I**) To show three-rowed membranelle 3 (arrow). (**J**) To show membranelle 2a (arrow), membranelle 2b (arrowhead), and paroral membrane (double arrowheads). (**K**) Left ventrolateral view of the holotype specimen. Ma, macronucleus. Scale bars = 50 μm.

#### 3.2.4. SSU rDNA Sequence

The SSU rDNA sequence of the Ningbo population of *Pleuronema orientale* has been deposited in the GenBank database with the accession number, length, and G + C content as follows: OQ591739, 1606 bp, 43.71%.

### 3.3. SSU rDNA Sequences and Phylogenetic Analyses

Phylogenetic trees inferred from SSU rDNA sequences using BI and ML analyses generated similar topologies. Therefore, only the ML tree is presented here, with supporting values from both algorithms (Figure 6).

Two newly sequenced species fall in the genus *Pleuronema*. The Ningbo population of *Pleuronema orientale* (OQ591739) clusters with *P. puytoraci* (KF840520) and the Hangzhou Bay population of *P. orientale* (KF206429) with full support (ML/BI, 100/1.00). *Pleuronema ningboensis* n. sp. (OQ591738) groups with the clade of *P. grolierei* (KF840519), *P. setigerum* (JX310015), *P. paucisaetosum* (KF206430) and *P.* cf. *setigerum* (FJ848875) with high support (ML/BI, 89/1.00), then the clade of the five species above clusters with the *P. coronatum* clade, but the support value is quite low (ML/BI, 22/0.73).

A BlastN analysis of two new sequences against the NCBI database is as follows. The SSU rDNA sequence of *Pleuronema ningboensis* n. sp. (OQ591738) is most similar to those of *P. paucisaetosum* (KF206430), *P. setigerum* (JX310015) and *P. grolierei* (KF840519), with 19–29 unmatched nucleotides and sequence identities 98.08–98.74% (Figure 7). The Ningbo population of *Pleuronema orientale* (OQ591739) differs from the Hangzhou Bay population (KF206429) and *P. puytoraci* (KF840520) in 5 and 2 nucleotides, respectively.

## 4. Discussion

The genus *Pleuronema* was established by Dujardin in 1841. It is a speciose and cosmopolitan genus comprising nearly 40 nominal species found in various aquatic environments [23,24,25,26,37,38]. The living cell size, ciliature, and especially the structure of the oral apparatus are the most important criteria for species separation within the genus [4,20]. Based on the structure of M2a, Wang et al. suggested that members of *Pleuronema* could be divided into two groups: ‘*coronatum*-type’ with the posterior end of M2a hook-shaped, ‘*marinum*-type’ characterized by the straight posterior end of M2a [39]. This view was widely accepted by later researchers and largely supported by the phylogenetic tree (Figure 6) [26]. In the present study, in terms of the structure of oral apparatus, both species belong to the ‘*coronatum*-type’ group.

### 4.1. Comparison of Pleuronema ningboensis n. *sp*. with Related Congeners

In the previous studies, more than twenty species have hook-shaped M2a and belong to the group of ‘*coronatum*-type’. Some of them can be clearly distinguished from *Pleuronema ningboensis* n. sp. according to the significant morphological differences, for example, *Pleuronema binucleatum* Pan et al., 2016; *P. lynni* Fernandez-Leborans & Novillo, 1993, *P. arctica* Agatha et al., 1993, and *P. salmastra* Dragesco & Dragesco-Kernéis, 1986 having more somatic kineties (at least 32 vs. 16–22 in *P. ningboensis* n. sp.) and preoral kineties (at least 6 vs. 3–5 in *P. ningboensis* n. sp.). *Pleuronema glaciale* Corliss & Snyder, 1986 with two rows of basal bodies M3 (vs. three rows in *P. ningboensis* n. sp.), *P. puytoraci* Groliere & Detcheva, 1974, *P. orientale* Pan et al., 2015 and *P. paraorientale* Liu et al., 2022 possessing larger cell size (70–120 × 45–70 μm, 95–135 × 50–85 μm, 95–115 × 55–70 μm, respectively, vs. 55–65 × 25–30 μm in *P. ningboensis* n. sp.) and more somatic kineties (28–29, 42–50, 52–62, respectively, vs. 16–22 in *P. ningboensis* n. sp.) [24,25,26,40,41,42,43,44,45].

According to the morphological characteristics of the new species, that is, the body shape (right ventrolateral straight), cell size (55–65 × 25–30 μm), number of somatic kineties (16–22), single spherical macronucleus, and the structure of oral apparatus (membrane 2a hook-like in posterior portion, two-rowed in anterior and posterior portions, single-rowed in middle portion; both membrane 1 and 3 three-rowed), five similar species should be compared with *Pleuronema ningboensis* n. sp., namely *P. paucisaetosum* Pan et al., 2015, *P. coronatum* Kent, 1881, *P. smalli* Dragesco, 1968, *P. foissneri* Liu et al., 2022, *P. parasmalli* Liu et al., 2022 (Table 2).

*Pleuronema coronatum* is one of the most common and well-studied scuticociliate species, which has been described by several times [6,20,39,46,47,48,49,50,51]. The body length in vivo of two Qingdao populations of *P. coronatum* described by Wang et al. [39] ranges from 55 to 170 μm. He proposed *Pleuronema balli*, *P. borrori* and *P. smalli* as synonyms of *P. coronatum* since they share some similar characteristics [20,39,46,48,49]. In the latest study, Liu et al. [26] recognized that only *Pleuronema balli* is a synonym of *P. coronatum* but regarded *Pleuronema borrori* and *P. smalli* as valid species based on body size, ciliature pattern, and macronucleus shape [26]. Therefore, based on the new reinvestigation, an improved diagnosis of *P. coronatum* was provided by Liu et al. [26].

*Pleuronema ningboensis* n. sp. can be obviously separated from *P. coronatum* by having fewer somatic kineties (16–22 vs. 28–48) and smaller body size (55–65 × 25–30 μm vs. 60–125 × 30–60 μm). In addition, they have different patterns of M2a in mid-portion arrangement (single-rowed in *P. ningboensis* n. sp. vs. zig-zag pattern in *P. coronatum*). A comparison of SSU rDNA sequences of *Pleuronema ningboensis* n. sp. and four *P. coronatum* shows that the unmatched nucleotides are 49–87 (sequence identities are 94.6–96.8%) [20,26,39,46].

*Pleuronema ningboensis* n. sp. closely resembles *P. paucisaetosum* in their small living cell size, similar ciliature pattern, and habitat. The former, however, can be distinguished from the latter by having different patterns of M3 (leftmost row shorter than other rows vs. three-rowed basically equal in length in *P. paucisaetosum*) and fewer somatic kineties (16–22. vs. 21–23 in *P. paucisaetosum*) [24]. The SSU rDNA sequence of *P. ningboensis* n. sp. is different from *P. paucisaetosum* (KF206430) by 19 bp with a sequence identity of 98.7% (Figure 7).

*Pleuronema smalli* was originally described by Dragesco [46] with the cell size after silver staining and ciliature pattern, but lack of morphological data of living cells. In terms of its small body size after silver staining and the number of preoral kineties, *Pleuronema ningboensis* n. sp. closely resembles the species *P. smalli*. The former, nevertheless, is different from the latter mainly by having fewer somatic kineties (16–22 vs. 28–36 in *P. smalli*) [26,46].

*Pleuronema ningboensis* n. sp. resembles *P. foissneri* in living cell size and the brackish water habitat; however, the former differs from the latter by fewer kineties (16–22 vs. 32–40 in *P. foissneri*), fewer preoral kineties (3–5 vs. 4–8 in *P. foissneri*) and different patterns of M2a arrangement in mid-portion (single-rowed vs. zig-zag pattern in *P. foissneri*). Meanwhile, *P. ningboensis* n. sp. has 87 different nucleotides (sequence identity is 94.6%) when compared with the sequence of *P. foissneri* (OL654416) [26].

*Pleuronema ningboensis* n. sp. has a similar body size, shape, and number of preoral kineties to *P. parasmalli*, but the former can be distinguished from the latter by possessing fewer somatic kineties (16–22 vs. 26–32), its brackish water habitat (vs. freshwater habitat) and different patterns of M2a mid-portion arrangement (single-rowed vs. zig-zag pattern). Comparing the SSU rDNA sequence of *P. ningboensis* n. sp. (OQ591738) and *P. parasmalli* (OL654417), the number of unmatched nucleotides is 104 (sequence identity is 93.9%) [26].

### 4.2. Comparison of Two Populations of Pleuronema orientale Pan et al., 2015

*Pleuronema orientale* was originally collected from a brackish water ditch of a marine island of Hangzhou Bay, China, and detailly described with morphology and molecular information [24]. The Ningbo population is consistent with the original description in most of the morphological features, such as the structure of the oral apparatus and the size of the living cell. There are some minor differences between the two populations, that is, the number of somatic kineties (36–51 in the present study vs. 42–50 in the original population), the number of preoral kineties (1–5 in the present study vs. 2–3 in the original population), and the number of macronuclei (1–3 in present study vs. 1 or 4 in the original population). In addition, the SSU rDNA sequence of the Ningbo population differs from the original population by only five unmatched nucleotides (99.7% in sequence identity). Consequently, we consider these variations to be population-dependent, and these two forms are conspecific.

As described above, two SSU rDNA sequences of *Pleuronema orientale* (OQ591739, KF206429) are very similar to *P. puytoraci* (KF840520), with two and three unmatched nucleotides, respectively. However, the morphological characteristics of *P. orientale* are quite different from the latter; that is, the former has more somatic kineties (36–51 in *P. orientale* vs. 28 or 29 in *P. puytoraci*), clearly separating them as two valid species [45,52]. *Pleuronema puytoraci* was originally reported by Grolière and Detcheva [52] with morphological and morphogenesis information. Pan et al. [45] provided detailed living characteristics and ciliary patterns based on a Hong Kong population, but unfortunately, they did not supply corresponding molecular data. Several years later, they supplied the SSU rDNA sequence of this species from another population without morphological information [23]. Considering the lack of corresponding morphological and molecular data from the same population, we doubt whether the current molecular information of *P. puytoraci* (KF840520) is correct. The morphological and molecular information of this species should be supplied based on the same population in the future.

### 4.3. Phylogenetic Relationships between Pleuronema and Schizocalyptra

The study of polyphyly of genus *Pleuronema* caused by *Schizocalyptra* nest within it was first discovered by Yi et al. [53]. This view was widely confirmed in most phylogenetic research based on the SSU rDNA sequence [23,24,25,26,54,55,56,57,58]. However, the phylogenetic analyses based on nuclear or mitochondrial data showed that the genus *Schizocalyptra* falls outside of *Pleuronema* [59,60]. In fact, the cell shape, the well-developed oral region with long paroral membranes, and three distinct membranelles M1–3 of *Schizocalyptra* are quite similar to that of *Pleuronema* [43]. It indicates that species in the genera *Schizocalyptra* and *Pleuronema* most likely evolved from a common ancestor. The present phylogenetic tree shows that *Schizocalyptra* nest within *Pleuronema*, consistent with previous studies based on SSU rDNA sequence data (Figure 6). However, the support valve is low, and the phylogenetic position of *Schizocalyptra* is unstable. Given the low and inconsistent support for the phylogenetic position of the genera *Schizocalyptra* and *Pleuronema* in previous and existing studies, more morphological and molecular information is needed to further clarify their positions.

## Figures and Tables

**Figure 6 microorganisms-11-01422-f006:**
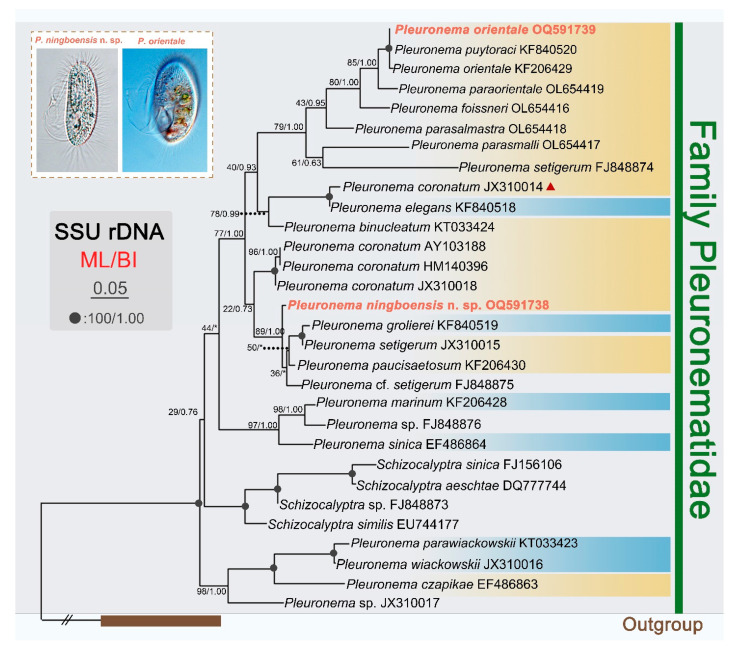
Maximum likelihood (ML) tree based on SSU rDNA sequences showing positions of *Pleuronema ningboensis* n. sp. and the Ningbo population of *Pleuronema orientale* (sequences in red). Numbers at nodes indicate the bootstrap values of maximum likelihood (ML) out of 1000 replicates and the posterior probabilities of Bayesian analysis (BI). Solid circles represent full bootstrap supports from both algorithms. Asterisks (*) indicate a mismatch in topologies between ML and BI analyses. The scale bar corresponds to five substitutions per 100 nucleotide positions. All branches are drawn to scale. The systematic classification mainly follows Gao et al. [28]. *Pleuronema coronatum* (JX310014) (marked with red triangle) deviates from the other three *P. coronatum* sequences; unfortunately, available information is not sufficient to determine its identity. *Pleuronema* clades of ‘*coronatum*-type’ and ‘*marinum*-type’ groups are marked with yellow and blue background colors, respectively.

**Figure 7 microorganisms-11-01422-f007:**
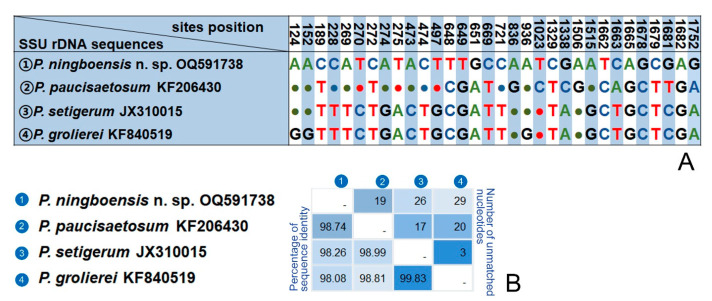
Sequence comparison of the SSU rDNA showing the unmatched nucleotides between *Pleuronema ningboensis* n. sp. and its most closely related congeners. (**A**) Nucleotide positions are given at the top of each column. Matched sites are represented by dots (·). (**B**) Matrix showing the percentage of sequence identity (below the diagonal) and the number of unmatched nucleotides (above the diagonal).

**Table 1 microorganisms-11-01422-t001:** Morphometric data of *Pleuronema ningboensis* n. sp. (first line) and *P. orientale* (second line) based on protargol staining specimen.

Character	Min	Max	Mean	Median	SD	CV (%)	N
Body length (μm)	47	64	54.5	54	5.08	9.32	25
	68	111	91.3	90	8.92	9.77	25
Body width (μm)	21	33	27.0	27	3.20	11.85	25
	41	77	56.2	58	7.86	13.99	25
Oral length (μm)	37	47	40.8	40	2.85	6.99	25
	53	84	68.3	67	5.69	8.33	25
Oral width (μm)	11	16	13.6	13	1.35	9.93	25
	16	30	21.9	22	2.62	11.96	25
Oral length/body length	0.61	0.87	0.75	0.75	0.06	8.00	25
	0.66	0.87	0.75	0.75	0.05	6.67	25
Number of somatic kineties	16	22	18.7	19	1.65	8.82	25
	36	51	46.6	47	3.04	6.52	25
Number of preoral kineties	3	5	4.1	4	0.76	18.54	25
	1	5	2.3	2	0.79	34.35	25
Number of kineties rows in M3	3	3	3.0	3	0	0	22
	3	3	3.0	3	0	0	25
Number of Ma	1	1	1.0	1	0	0	25
	1	3	1.2	1	0.50	41.67	25
Diameter of macronucleus (μm)	18	31	23.9	23	3.50	14.64	25
	21	67	32.2	31	1.68	5.22	25

All data based on protargol-impregnated specimens. Abbreviations: CV, coefficient of variation in %; M, Median; Max, maximum; Mean, arithmetic mean; Min, minimum; N, number of specimens investigated; SD, standard deviation.

**Table 2 microorganisms-11-01422-t002:** Morphological comparison of *Pleuronema ningboensis* n. sp. with related congeners.

Species	*P. ningboensis*	*P. paucisaetosum*	*P. coronatum*	*P. smalli*	*P. foissneri*	*P. parasmalli*
Body size in vivo (μm)	55–65 × 25–30	55–85 × 25–55	60–125 × 30–60	-	60–75 × 30–40	55–85 × 25–35
Body size after staining (μm)	47–64 × 21–33	60–80 × 36–58	50–115 × 25–65	49–70 × 28–41	60–90 × 35–50	60–80 × 30–40
Right ventrolateral side in vivo	Basically straight	-	Straight	-	Convex	Basically straight
Number of caudal cilia	10–17	12–15	10–15	-	About 15	15–18
Number of somatic kineties	16–22	21–23	35–48	28–36	32–40	26–32
Number of preoral kineties	3–5	4 or 5	3–7	2–5	4–8	4–6
Number of kinety rows in M3	3	3	3	2	3	3
Number of kinety rows in M1	3	3	2	-	3	3
Mid-portion of M2a	Single-rowed	Single-rowed	Zig-zag pattern	-	Zig-zag pattern	Zig-zag pattern
Habitat	Brackish water	Brackish water	Freshwater and marine water	Brackish and marine water	Brackish water	Freshwater
Data sources	Present study	[24]	[26]	[46]	[26]	[26]

Abbreviations: M1, membranelle 1; M2a, membranelle 2a; M3, membranelle 3; -, data not available.

## Data Availability

The data presented in the study are deposited in the NCBI database repository, accession numbers: OQ591738 and OQ591739.

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
