# Peer review of "Taxonomic and Phylogenetic Studies of Two Brackish Pleuronema Species (Protista, Ciliophora, Scuticociliatia) from Subtropical Coastal Waters of China, with Report of a New Species"

_microorganisms, 2023, doi:10.3390/microorganisms11061422_

Round 1
Reviewer 1 Report
This is a clear presentation of two species of an interesting group of ciliates. I don't see any deficiencies to bring up.
Author Response
Thank you for your time review our MS.
Reviewer 2 Report
The manuscript entitled as 'Taxonomic and phylogenetic studies of two brackish Pleuronema species (Protista, Ciliophora, Scuticociliatia) from subtropical coastal waters of China, with report of a new species' provides taxonomic descriptions based on modern techniques. The genus Pleuronema is common ciliates in coastal environments and a species-rich group. However, the species-rich might result from insufficient description of old species and narrower boundary for each species. Even though Zhang et al. provided better and more information about the two Chinese Pleuronema species and the DNA sequence would be a good criterion when one distinguishes between congeners, there are some discrepancies between drawings and descriptions. For instance, the sizes of contractile vacuole and macronucleus did not match. And anterior suture of somatic kineties is enigmatic. Additionally, if the DNA sequence came from only a single specimen without any duplicates, I could not convince there are no contamination. Unfortunately, Zhang et al. did not include a silver nitrate preparation which will give us the position and number of contractile vacuole pore. For minor issues, see below.
-27: Pleuronema is polyphyletic: Because of low supporting values for the clades (Fig. 6), we could not say it is polyphyletic.
-51: Pleuronema orientlae -. P. orientlae
-58: Samples were collected from subtropical brackish waters in Ningbo, China, which are several coastal wetlands near East China Sea (Figure 1). -> Samples of subtropical brackish waters were collected from coastal wetlands in Ningbo, China (Figure 1).
-83: single cell: Only a single cell was sequenced for each species? Without any duplicates? Could you confirm that it is not contaminated with closely-related other species?
-91: Phylogenetic analyses: Who are the outgroup? Provide the list.
-106: 106: There are something wrong, right? Want to say 1,000,000?
-119: shortened. -> shortened posteriorly.
-126: Type materials: It is highly recommended to deposit some materials to other national institute.
-131: Morphological description: Extrusomes are missing. They do not have the extrusomes? From Fig. 3D, the extrusomes was mentioned.
-138: contractile vacuole dorsally located: From micrographs, I did not confirm the position of CV. It could be located either dorsally or ventrolaterllay. Did you check the position of CV pore? Check the row 204 as well.
-140: A spherical macronucleus positioned one-third down length of body -> A spherical macronucleus positioned in body center slightly anteriorly.
-168: Single spherical to elliptical macronucleus, diameter about 24 μm after protargol staining (Figure 3J, Table 1): This is a section for somatic kineties. Remove it. Check the rows 215-217 as well.
-170: M1: Based on Fig. 3J. I can find a small ring structure made by posterior basal bodies of the M1 that can be observed in P. setigerum sensu Pan et al. (2015). Please describe it.
-251-253: The genus Pleuronema comprises four separate clades, and the members of the genus Schizocalyptra forms a full-support clade and placed within Pleuronema, making Pleuronema polyphyletic: The genus Schizocalyptra was nested in the Pleuronema clade, but the position of Schizocalyptra is not robust. See the clade including the Schizocalyptra and Pleuronema, it showed supporting values of 29/0.76, so that the grouping is meaningless.
-335: leftmost -> rightmost?
Fig. 2A: CV: The CV is too small. Based on the description, it should be 12-15 microns in diameter; Ma: As well, it is too small. Check your description and morphometrics.
Fig. 2B: Anterior suture of somatic kinoites: The drawing of somatic kineties might be wrong. The genus Pleuronema has the suture, but the 2B does not show an ordinary suture. The anterior somatic kineties should converge as a suture running along the apical barren area. Probably some anterior basal bodies of PK are those of right somatic kineties. See Fig. 3J.
The manuscript entitled as 'Taxonomic and phylogenetic studies of two brackish Pleuronema species (Protista, Ciliophora, Scuticociliatia) from subtropical coastal waters of China, with report of a new species' provides taxonomic descriptions based on modern techniques. The genus Pleuronema is common ciliates in coastal environments and a species-rich group. However, the species-rich might result from insufficient description of old species and narrower boundary for each species. Even though Zhang et al. provided better and more information about the two Chinese Pleuronema species and the DNA sequence would be a good criterion when one distinguishes between congeners, there are some discrepancies between drawings and descriptions. For instance, the sizes of contractile vacuole and macronucleus did not match. And anterior suture of somatic kineties is enigmatic. Additionally, if the DNA sequence came from only a single specimen without any duplicates, I could not convince there are no contamination. Unfortunately, Zhang et al. did not include a silver nitrate preparation which will give us the position and number of contractile vacuole pore. For minor issues, see below.
-27: Pleuronema is polyphyletic: Because of low supporting values for the clades (Fig. 6), we could not say it is polyphyletic.
-51: Pleuronema orientlae -. P. orientlae
-58: Samples were collected from subtropical brackish waters in Ningbo, China, which are several coastal wetlands near East China Sea (Figure 1). -> Samples of subtropical brackish waters were collected from coastal wetlands in Ningbo, China (Figure 1).
-83: single cell: Only a single cell was sequenced for each species? Without any duplicates? Could you confirm that it is not contaminated with closely-related other species?
-91: Phylogenetic analyses: Who are the outgroup? Provide the list.
-106: 106: There are something wrong, right? Want to say 1,000,000?
-119: shortened. -> shortened posteriorly.
-126: Type materials: It is highly recommended to deposit some materials to other national institute.
-131: Morphological description: Extrusomes are missing. They do not have the extrusomes? From Fig. 3D, the extrusomes was mentioned.
-138: contractile vacuole dorsally located: From micrographs, I did not confirm the position of CV. It could be located either dorsally or ventrolaterllay. Did you check the position of CV pore? Check the row 204 as well.
-140: A spherical macronucleus positioned one-third down length of body -> A spherical macronucleus positioned in body center slightly anteriorly.
-168: Single spherical to elliptical macronucleus, diameter about 24 μm after protargol staining (Figure 3J, Table 1): This is a section for somatic kineties. Remove it. Check the rows 215-217 as well.
-170: M1: Based on Fig. 3J. I can find a small ring structure made by posterior basal bodies of the M1 that can be observed in P. setigerum sensu Pan et al. (2015). Please describe it.
-251-253: The genus Pleuronema comprises four separate clades, and the members of the genus Schizocalyptra forms a full-support clade and placed within Pleuronema, making Pleuronema polyphyletic: The genus Schizocalyptra was nested in the Pleuronema clade, but the position of Schizocalyptra is not robust. See the clade including the Schizocalyptra and Pleuronema, it showed supporting values of 29/0.76, so that the grouping is meaningless.
-335: leftmost -> rightmost?
Fig. 2A: CV: The CV is too small. Based on the description, it should be 12-15 microns in diameter; Ma: As well, it is too small. Check your description and morphometrics.
Fig. 2B: Anterior suture of somatic kinoites: The drawing of somatic kineties might be wrong. The genus Pleuronema has the suture, but the 2B does not show an ordinary suture. The anterior somatic kineties should converge as a suture running along the apical barren area. Probably some anterior basal bodies of PK are those of right somatic kineties. See Fig. 3J.
Author Response
Below, please find our responses for your every useful comments.
The manuscript entitled as 'Taxonomic and phylogenetic studies of two brackish Pleuronema species (Protista, Ciliophora, Scuticociliatia) from subtropical coastal waters of China, with report of a new species' provides taxonomic descriptions based on modern techniques. The genus Pleuronema is common ciliates in coastal environments and a species-rich group. However, the species-rich might result from insufficient description of old species and narrower boundary for each species. Even though Zhang et al. provided better and more information about the two Chinese Pleuronema species and the DNA sequence would be a good criterion when one distinguishes between congeners, there are some discrepancies between drawings and descriptions. For instance, the sizes of contractile vacuole and macronucleus did not match. And anterior suture of somatic kineties is enigmatic. Additionally, if the DNA sequence came from only a single specimen without any duplicates, I could not convince there are no contamination. Unfortunately, Zhang et al. did not include a silver nitrate preparation which will give us the position and number of contractile vacuole pore. For minor issues, see below.
RE: Thanks for your comments and suggestions. All the issues have been modified. Please check the response information and the revised manuscript.
-27: Pleuronema is polyphyletic: Because of low supporting values for the clades (Fig. 6), we could not say it is polyphyletic.
RE: We have deleted the statement.
-51: Pleuronema orientlae -. P. orientlae
-58: Samples were collected from subtropical brackish waters in Ningbo, China, which are several coastal wetlands near East China Sea (Figure 1). -> Samples of subtropical brackish waters were collected from coastal wetlands in Ningbo, China (Figure 1).
-106: 106: There are something wrong, right? Want to say 1,000,000?
-140: A spherical macronucleus positioned one-third down length of body -> A spherical macronucleus positioned in body center slightly anteriorly.
-168: Single spherical to elliptical macronucleus, diameter about 24 μm after protargol staining (Figure 3J, Table 1): This is a section for somatic kineties. Remove it. Check the rows 215-217 as well.
RE: All the minor issues mentioned above have been modified.
-119: shortened. -> shortened posteriorly.
RE: Here we stated the leftmost row of M3, from the individual’s perspective (not the position in the photomicrographs). Therefore, we modified it as “leftmost row shortened and posteriorly located, rightmost row slightly shortened posteriorly”. Arrows/arrowheads were also added in Fig. 3I to indicate the leftmost and rightmost rows of M3.
-83: single cell: Only a single cell was sequenced for each species? Without any duplicates? Could you confirm that it is not contaminated with closely-related other species?
RE: For each species, about five to ten cells were placed into several Eppendorf tubes with one or multiple cells, and then sequenced separately. The sequence results were identical, which, in our opinion, could be identified as coming from the same species. We have added more details in Section 2.2.
-91: Phylogenetic analyses: Who are the outgroup? Provide the list.
RE: The outgroup includes five sequences, namely Falcicyclidium plouneouri (FJ868181), Falcicyclidium fangi (FJ868185), Acucyclidium atractodes (FJ868182), Hippocomos salinus (JX310012), and Wilbertia typica (FJ490551). The outgroup information was provided in the manuscript. We also provide a supplementary material (Table S1) with species names and GenBank accession numbers of all sequences included in the present phylogenetic analyses.
-126: Type materials: It is highly recommended to deposit some materials to other national institute.
RE: Thanks for your suggestion. Two slides with protargol-stained paratype specimens (registration number: YTT-20191123-2-3, 2-4) have been deposit to Laboratory of Protozoology, Ocean University of China, one of the central laboratories for ciliate diversity research. The details have been modified in the revised manuscript.
-131: Morphological description: Extrusomes are missing. They do not have the extrusomes? From Fig. 3D, the extrusomes was mentioned.
RE: We have the description of extrusomes in Section 3.1.5 of the revised manuscript Line 142–144 (line numbers that after showing all the tracking changes).
-138: contractile vacuole dorsally located: From micrographs, I did not confirm the position of CV. It could be located either dorsally or ventrolaterllay. Did you check the position of CV pore? Check the row 204 as well.
RE: According to Fig. 3D, the oral region is located at the left of the micrograph, which indicates the position of the ventral side of the cell. The contractile vacuole (Fig. 3D, arrowhead) is relatively apart from the ventral side (oral region) and more closely located to the dorsal side. Thus, we describe the contractile vacuole as dorsally located. CV pores were not observed in vivo or after staining, and silver nitrate preparation was not able to apply.
-170: M1: Based on Fig. 3J. I can find a small ring structure made by posterior basal bodies of the M1 that can be observed in P. setigerum sensu Pan et al. (2015). Please describe it.
RE: In Pan et al. (2015), a small ring structure was observed at the posterior end of M2a (Page 37, Fig. 3L). However, we examined all specimens of our new species and determined that posterior portion of M2a of Pleuronema ningboensis does not arrange in a ring-like structure. In Fig 3J, it is very likely to be caused by impurities after protargol staining. We also add a detailed description in Line 185 in the revised manuscript showing tracking changes.
-251-253: The genus Pleuronema comprises four separate clades, and the members of the genus Schizocalyptra forms a full-support clade and placed within Pleuronema, making Pleuronema polyphyletic: The genus Schizocalyptra was nested in the Pleuronema clade, but the position of Schizocalyptra is not robust. See the clade including the Schizocalyptra and Pleuronema, it showed supporting values of 29/0.76, so that the grouping is meaningless.
RE: We agree that it is premature to state Pleuronema to be polyphyletic due to the low supporting values of the position of Schizocalyptra. In the revised manuscript, we have deleted the related statements.
-335: leftmost -> rightmost?
RE: We have confirmed that it is the leftmost row of M3 (in Fig. 3I, the shorter row of M3 is present at the right side, away from M2b, but from the individual’s perspective, it is the leftmost row of M3 since Fig. 3I shows its ventral side). Arrowheads were also added in Fig. 3I to indicate the leftmost and rightmost rows of M3.
Fig. 2A: CV: The CV is too small. Based on the description, it should be 12-15 microns in diameter; Ma: As well, it is too small. Check your description and morphometrics.
Fig. 2B: Anterior suture of somatic kinoites: The drawing of somatic kineties might be wrong. The genus Pleuronema has the suture, but the 2B does not show an ordinary suture. The anterior somatic kineties should converge as a suture running along the apical barren area. Probably some anterior basal bodies of PK are those of right somatic kineties. See Fig. 3J.
RE: Figure 2 has been modified based on these comments.